# Genetic Bias, Diversity Indices, Physiochemical Properties and CDR3 Motifs Divide Auto-Reactive from Allo-Reactive T-Cell Repertoires

**DOI:** 10.3390/ijms22041625

**Published:** 2021-02-05

**Authors:** Oscar L. Haigh, Emma J. Grant, Thi H. O. Nguyen, Katherine Kedzierska, Matt A. Field, John J. Miles

**Affiliations:** 1Center for Immunology of Viral, Auto-Immune, Hematological and Bacterial Diseases (IMVA-HB), Commissariat à l’Énergie Atomique et aux Énergies renouvelables, Université Paris-Saclay, INSERM U1184, Fontenay-aux-Roses, France; oscar.haigh@cea.fr; 2Department of Biochemistry and Genetics, La Trobe Institute for Molecular Science, La Trobe University, Melbourne, VIC 3086, Australia; E.Grant@latrobe.edu.au; 3Department of Microbiology and Immunology, at the Peter Doherty Institute for Infection and Immunity, University of Melbourne, Parkville, VIC 3010, Australia; tho.nguyen@unimelb.edu.au (T.H.O.N.); kkedz@unimelb.edu.au (K.K.); 4The Australian Institute of Tropical Health and Medicine (AITHM), James Cook University, Cairns, QLD 4811, Australia; matt.field@jcu.edu.au; 5Centre for Molecular Therapeutics, James Cook University, Cairns, QLD 4811, Australia; 6Centre for Tropical Bioinformatics and Molecular Biology, James Cook University, Cairns, QLD 4811, Australia; 7John Curtin School of Medical Research, Australian National University, Canberra, ACT 2601, Australia; 8QIMR Berghofer Medical Research Institute, Brisbane, QLD 4006, Australia

**Keywords:** T-cell, T-cell receptor, T-cell repertoire, T-cell alloreactivity

## Abstract

The distinct properties of allo-reactive T-cell repertoires are not well understood. To investigate whether auto-reactive and allo-reactive T-cell repertoires encoded distinct properties, we used dextramer enumeration, enrichment, single-cell T-cell receptor (TCR) sequencing and multiparameter analysis. We found auto-reactive and allo-reactive T-cells differed in mean ex vivo frequency which was antigen dependent. Allo-reactive T-cells showed clear differences in TCR architecture, with enriched usage of specific T-cell receptor variable (*TRBJ*) genes and broader use of T-cell receptor variable joining (*TRBJ*) genes. Auto-reactive T-cell repertoires exhibited complementary determining regions three (CDR3) lengths using a Gaussian distribution whereas allo-reactive T-cell repertoires exhibited distorted patterns in CDR3 length. CDR3 loops from allo-reactive T-cells showed distinct physical-chemical properties, tending to encode loops that were more acidic in charge. Allo-reactive T-cell repertoires differed in diversity metrics, tending to show increased overall diversity and increased homogeneity between repertoires. Motif analysis of CDR3 loops showed allo-reactive T-cell repertoires differed in motif preference which included broader motif use. Collectively, these data conclude that allo-reactive T-cell repertoires are indeed different to auto-reactive repertoires and provide tangible metrics for further investigations and validation. Given that the antigens studied here are overexpressed on multiple cancers and that allo-reactive TCRs often show increased ligand affinity, this new TCR bank also has translational potential for adoptive cell therapy, soluble TCR-based therapy and rational TCR design.

## 1. Introduction

Many questions remain on the distinct properties of allo-reactive T-cells [1]. Despite the fact that allo-reactive T-cells are an established clinical correlate of graft rejection, knowledge of the precise molecular characteristics of this T-cell repertoire are largely unknown. Initial T-cell receptor (TCR) repertoire studies have shown TCR repertoire structure correlates with rejection [2,3], cancer protection post-transplant [4], pathogen infection post-transplant [3], and high TCR avidity [5].

Three models of alloreactivity have been proposed [6]. First, is that allo-reactive TCRs recognise foreign human leukocyte antigen (HLA) molecules directly, with the peptide (p) being irrelevant, termed the “promiscuous model”. Second, is that allo-reactive TCRs rearrange their CDR3 loops to recognise promiscuous shapes, termed the “degenerate model”. Third, is that allo-reactive TCRs bind both peptide and HLA in conventional form, making small changes in the contact interface that dramatically modify binding affinity, termed the “polyspecific model”. Massing functional, biophysical, and structural evidence supports the polyspecific model with allo-reactive TCRs binding both HLA and peptide in a conventional manner, often through molecular mimicry binding “hotspot” regions [7,8,9,10].

Solid-organ transplant (SOT) and hematopoietic stem-cell transplantation (HSCT) rejection can be controlled using T-cell suppressing drugs such as cyclosporine, corticosteroids, rapamycin or infusions with T-cell suppressing monoclonal antibodies. However, these life-long T-cell suppressing drug regiments come with increased risk of infection, cardiovascular disease and cancer [11]. Acute graft rejection may occur between one-week and three-weeks post-transplant. Chronic rejection can occur over many years. It is thought that chronic rejection can be triggered by innocuous infections, which activate allo-reactive memory T-cells that damage host tissue [12]. Many SOT rejection episodes involve T cell-mediated damage, but these episodes do not always destroy grafts [13,14].

T-cell mediated HSCT rejection comprises three modalities. First, T-cells can recognise host pHLA directly, termed the “direct alloresponse”. Second, T-cell-derived peptide fragments presented by donor HLA can be recognised, termed the “indirect alloresponse”. Third, T-cells can recognise pMHC molecules that have been transferred from donor cells to recipient cells, termed the “semidirect alloresponse” [14].

Currently, structural, biophysical, mutational, and functional analysis of auto-reactive and allo-reactive TCRs and pHLAs agrees with a conventional top-down docking mode, where both the germline loops of the TCR and the somatic complementary determining regions three (CDR3) loops contact both the peptide and HLA. This docking occurs in a roughly diagonal manner with binding affinities between K_D_ = 0.3 µM and above 1 mM [9,10,15,16]. However, this conventional docking mode can sometimes vary [17]. The mouse major histocompatibility complex (MHC-I) restricted 2C TCR binds its auto-reactive ligand at K_D_ = 83–85 µM, approximately 40-fold weaker its allo-reactive ligands at K_D_ = 2–3 µM [10,18,19]. In humans, the antiviral LC13 TCR binds its auto-reactive ligand at K_D_ = 10 µM [16], approximately 10-fold weaker than its allo-reactive ligand at K_D_ = 1 µM [17]. The structures of these complexes show they bind both HLA and peptide with conventional footprints [17,18,19,20,21,22,23]. Of additional interest is the auto-reactive MHC-II-restricted mouse TCR YAe62 can also bind a MHC-I ligand [24]. This TCR transcends MHC-restriction dogma. The same phenomenon has been shown in humans, with a virus-specific HLA-I-restricted TCR being allo-reactive with an HLA-II-restricted ligand [25]. TCRs are also known to recognise different antigens presented by different HLA-I alleles [26,27]. In general, the polyspecific model appears to best conform with alloreactivity [22], with the peptide being the focus of affinity variation [28] through increased atomic contacts [22]. However, TCR/pMHC structural correlations or multimer correlations have not yet been explicitly linked with graft acceptance or rejection in humans.

In this study, we aimed to investigate two gaps in knowledge. First, was to determine whether antigen-specific precursors differed in frequency between auto-reactive and allo-reactive individuals. Second, was to determine if T-cell repertoires could be separated into auto-reactive and allo-reactive groups when quantitating TCR gene preference, TCR gene pairings, CDR3 length, CDR3 physiochemical-properties, CDR3 diversity metrics and CDR3 motifs. Here, we used HLA-0201 dextramers bearing four different immunogenic tumour associated antigens (TAAs). The first was IMNDMPIYM (herein referred to as IMN) derived from the transmembrane receptor EphA2 and overexpressed in prostate, breast, colon, gastric cancers, and metastatic melanoma [29,30,31]. Second, was FLCMKALLL (herein referred to as FLC) [32] derived from the android receptor ligand-binding domain protein and overexpressed in prostate cancer, breast cancers and hepatocellular carcinoma [33,34,35]. Third, was KVAELVHFL (herein referred to as KVA) derived from the MAGE-A3 protein and overexpressed in prostate, breast, and esophageal cancers [36,37,38]. Fourth, was YLEPGPVTV (herein referred to as YLE) derived from the gp100 protein, which is found in premelanosomal pigmented cells and the retina and overexpressed in numerous cancers, including prostate cancer [38]. In this study, HLA-0201^+^ donors are referred to as auto-reactive and HLA-0201^−^ donors are referred to as allo-reactive.

## 2. Results

### 2.1. Antigen-Specific T-Cells Show Different Frequencies between Auto-Reactive and Allo-Reactive Settings

Previous experience enumerating and sorting extremely low-frequency T-cell populations in cord and adult blood [39] has shown us success requires; (i) the use of a bright fluorochrome on the dextramer [40]; (ii) the use of freshly made dextramers which increases brightness 13-to-50-fold, compared with other multimers [39,41]); (iii) the use of the protein kinase inhibitor dasatinib which prevents TCR internalisation, activation-induced cell death [42] and increases multimer brightness up to 50-fold [40]; (iv) cell dumping using LIVE/DEAD, CD14, CD16 and CD19 [40] and; (v) a decreased sorter flow rate which minimises the abort rate of dextramer+ cells. An important variable in this protocol is the functional avidity of the T-cell population under investigation, which derives from the physiology and activation status of the cells.

Using flow cytometry, we found that IMN- and FLC-specific T-cells had variable mean frequencies across auto-reactive and allo-reactive populations (Figure 1A,B), YLE-specific T-cells had a significantly higher mean frequency (*p* ≤ 0.0002) in auto-reactive populations than allo-reactive populations (Figure 1C). KVA-specific T-cells exhibited a higher mean frequency when comparing auto-reactive and allo-reactive populations (Figure 1D). Collectively, the frequency hierarchy was FLC > IMN > KVA > YLE, ranging from a mean of 0.004−0.06% of CD8^+^ T-cells. Examples of dextramer plots from ex vivo samples are shown (Figure 1E–H), revealing the low percentages of these cells in the blood. Examples of samples that have been sorted twice for purity (and further single-cell sorting) are also shown (Figure 1I,J), highlighting that antigen-specific T-cells can be reliably isolated and enriched from both auto-reactive and allo-reactive settings.

### 2.2. The IMN Allo-Reactive T-Cell Repertoire Exhibits Expanded TRBV Gene Usage and Unconventional CDR3 Loop Length 

We next performed single-cell sorting and TCR analysis on IMN- and FLC-specific T-cells given that these dextramers had the highest mean fluorescence intensity (MFI) of the dextramer set and had the highest frequencies in the blood. These features help confirm precise sorting and provided confidence in downstream analysis. IMN-specific TCR sequences were then pooled and normalised by clonotype count and appearance. Differences in TRBV usage was highly significant between cohorts showing 77% of total variation by two-way ANOVA (*p* ≤ 0.0001) (Figure 2A). Here, 27-of-30 (90%) TRBV genes were deployed in allo-reactive repertoires versus 19-of-30 (63%) in auto-reactive repertoires. The total variation in *TRBJ* usage was also highly significant between cohorts at 92% by two-way ANOVA (*p* ≤ 0.0001) (Figure 2B). There was also a distinct difference in CDR3 length between repertoires with the allo-reactive repertoire not exhibiting a conventional Gaussian distribution in CDR3 length, which typically shows a peak at 15 residues in length. The CDR3 length variation between cohorts was 76% and significant by two-way ANOVA (*p* ≤ 0.025) (Figure 2C). Interestingly, allo-reactive TCRs tended to encode shorter CDR3 loops. Here, in allo-reactive CDR3 loops, 61% of the repertoire was under 13 residues in length versus 18% in the auto-reactive cohort. There were no significant differences in TRBD usage between auto-reactive (Figure 2D) and allo-reactive (Figure 2E) TCRs, with the TRBD1*01 gene predominating in both cohorts.

We then compared total clonotype frequencies in three auto-reactive donors (Figure 2F) and three allo-reactive donors (Figure 2G) by total abundance sampling. Here, three-of-three auto-reactive donors exhibited multiple copies of an individual clonotype against IMN, with frequencies up to 93% for a single clonotype in donor 1 (Figure 2F). Within the allo-reactive donors, one-of-three individuals exhibited multiple copies of an individual clonotype against IMN, with a frequency of 16% for a single clonotype in donor 4 (Figure 2G).

### 2.3. IMN-Specific T-Cell Repertoires Exhibit Divergent TCR Gene Pairing

IMN-specific T-cell repertoires were next compared for TRBV/*JTRBJ* and TRAV/TRAJ pairing bias using normalised counts by abundance and chord diagram analysis (Figure 3). In the auto-reactive repertoire, the large unique multiple copy clonotype from auto-reactive donor 1 (TRAV12-2/TRBV5-4) was observed along with preferential pairing of TRBV7-9 and *JTRBJ*2-1 for *TRB* genes and TRAV12-2 and TRAJ27 for the TRA genes. TCR gene pairings in the allo-reactive repertoire were much more complex. However preferential pairings of TRBV7-9 and TRBV2 with *JTRBJ*2-7 were observed for the TRB genes and TRAV12-2 and TRAJ39 for the TRV genes. More TRAJ genes were utilized in the allo-reactive repertoire compared with the auto-reactive repertoire.

### 2.4. The FLC-Specific Allo-Reactive T-Cell Repertoire Exhibits Divergent TRBV and TRBJ Gene Usage, Unconventional CDR3 Loop Length and Common Multiple Copy Clonotypes

Like the IMN-specific repertoires, FLC-specific TCR sequences were also pooled and normalised by clonotype count and appearance. Differences in TRBV usage were highly significant showing 73.86% between cohorts by two-way ANOVA (*p* ≤ 0.005) (Figure 4A). Differences in *TRBJ* usage were also significant showing 80% variation by two-way ANOVA (*p* ≤ 0.016) (Figure 4B). Akin to allo-reactive IMN repertoire, the FLC-specific allo-reactive repertoire did not exhibit a Gaussian distribution in CDR3 length showing 90% variation compared to the auto-reactive repertoire by two-way ANOVA (*p* ≤ 0.0001) (Figure 4C). Allo-reactive CDR3 sequences were shorter than auto-reactive sequences but not to the same degree as seen in the IMN system, with 37% and 34% of total sequences, respectively being under 13 residues in length. There were no significant differences in TRBD usage between auto-reactive (Figure 4D) and allo-reactive (Figure 4E) TCRs, with the TRBD1*01 gene predominating in both cohorts.

When examining the total clonotype frequencies per donor, two-of-three auto-reactive donors showed unique clonotypes with multiple copies (Figure 4F) versus three-of-three in allo-reactive donors (Figure 4G). Donor 12 showed the largest unique clonotype frequency of 46%. These multiple copy clonotypes in FLC allo-reactive donors are different to the IMN system which were comprised of single unique clonotypes per donor.

### 2.5. FLC-Specific T-Cell Repertoires Exhibit Divergent TCR Gene Pairing

FLC-specific T-cell repertoires were next compared for TRBV/*TRBJ* and TRAV/TRAJ pairing bias using normalised counts by abundance and chord diagram analysis (Figure 5). In the auto-reactive repertoire, we observed preferential pairing of TRBV20-1 and *TRBJ*2-5 for TRB genes and TRAV5 and TRAJ47 for the TRA genes. TCR gene pairings in the allo-reactive repertoire were again more complex. However, preferential pairings of TRBV19 and TRBV2-1 were observed for the TRB genes and TRAV19 and TRAJ49 for the TRV genes. Akin to the IMN system, more TRAJ genes were utilized in the allo-reactive repertoire compared with the auto-reactive repertoire.

### 2.6. T-Cells Exhibit Divergent CDR3 Physical-Chemical Properties between Auto-Reactive and Allo-Reactive Repertoires

We next examined the physical-chemical properties of the CDR3 loops in the IMN- and FLC-specific repertoires by pooling the sequences based on appearance. In both systems, all metrics measured were different between auto-reactive and allo-reactive repertoires (Figure 6) and combined, showed 95% variance by two-way ANOVA (*p* ≤ 0.0001). In the IMN cohorts, auto-reactive CDR3 were more hydrophobic (*p* ≤ 0.04) and had higher mean grand average of hydropathicity index (GRAVY) scores showing these receptors had a low dielectric constant to water. Auto-reactive IMN CDR3 also had higher average molecular weights (Mw) (*p* = 0.008) and average monoisotopic Mw (*p* ≤ 0.008), showing that these CDR3 were longer and/or have bulkier side chains. Auto-reactive IMN TCR also had higher mean theoretical pI suggesting that these receptors were more basic in charge. Interestingly, this pattern of hydrophobic, GRAVY, Mw and monoisotopic Mw metrics was inverted in the FLC system, showing that the physical-chemical properties of CDR3 loops are largely determined by antigen. Of note, was an increase in mean in theoretical pI in auto-reactive FLC-specific receptors which matched the IMN system.

### 2.7. T-Cells Exhibit Divergent CDR3 Diversity Metrics between Auto-Reactive and Allo-Reactive Repertoires

We next quantitated and compared diversity metrics of IMN- and FLC-specific CDR3 loops from individual repertoires from auto-reactive and allo-reactive donors (Table 1). While individual diversity metrics did not reach significance, likely due to the low numbers of donors in each cohort (*n* = 3), the combined metrics explained 45% (*p* ≤ 0.0008) and 50% (*p* ≤ 0.0001) of the variation between the IMN and FLC cohorts, respectively. Notably, the same Log_2_ fold-change (Log_2_FC) patterns were observed in both IMN and FLC systems suggesting these measures may be valid on an expanded donor analysis. The most heavily weighted diversity measurements are shown, as are the Log_2_FC changes with increases shown in blue decreases shown in red.

Using 12 formulas of diversity, we compared each system. Both IMN and FLC repertoires showed that allo-reactive TCR had increased numbers of unique clonotypes per individual (Menhinick index and Margalef Richness index), showed fewer multiple copy clonotypes but when they were present, were more prevalent in the repertoire (Dominance index and Gini coefficient). Allo-reactive TCR showed an overall increase in diversity which factors both total unique clonotypes and unique clonotypes with multiple copy numbers (Simpson’s index, Reciprocal Simpson index, Berger-Parker Dominance index, and Inverted Berger-Parker Dominance index) and this diversity was more homogenous between individuals (Buzas and Gibson’s index, Shannon index and Equitability index).

### 2.8. T-Cells Exhibit Divergent CDR3 Motifs between Auto-Reactive and Allo-Reactive Repertoires

Finally, we examined CDR3 residue composition using software that detects common CDRR3 motifs from a large database [43] (Figure 7). 118 CDR3 motifs were identified in the IMN repertoire and variation between the auto-reactive and allo-reactive repertoires was 84% by two-way ANOVA (*p* ≤ 0.0001). The allo-reactive repertoire contained more CDR3 motifs than the auto-reactive repertoire at 93% and 84%, respectively. 124 CDR3 motifs were identified in the IMN repertoire and variation between the auto-reactive and allo-reactive repertoires was 63% by two-way ANOVA (*p* ≤ 0.0013). Akin to the IMN system. The allo-reactive repertoire contained more CDR3 motifs than the auto-reactive repertoire at 87% and 80%, respectively. Interestingly, the YGQ motif was identified only in IMN and FLC allo-reactive repertoires. These data show that the TCR repertoires of IMN- and FLC-specific T-cells show different structural architecture between auto-reactive and allo-reactive individuals.

## 3. Discussion

The distinct properties of allo-specific T-cell repertoires are not well known. This study used dextramers, single-cell TCR analysis and multiple annotation analysis to determine if alloreactive TCR repertoires could be identified from auto-reactive TCR repertoires. The enumeration of T-cells from ex vivo blood showed variability between antigens (IMN, FLC, YLE and KVA), with low mean frequencies, varying from 0.004 to 0.06% of CD8^+^ T-cells even among auto-reactive donors. This was expected, as self-specific T-cells are typically low-frequency, apart from the abnormal MART-1 antigen which can have frequencies up to 0.35% of CD8^+^ T-cells ex vivo [41], due to a unique germline focused docking mode [44,45]. Importantly, dextramer enumeration across all four antigens showed a consistent difference in mean frequency when comparing auto-reactive T-cells to allo-reactive T-cells.

Surprisingly, single-cell TCR analysis revealed multiple copies of unique clonotypes in some allo-reactive populations, suggesting possible cross-reactivity with a pre-existing memory T-cell response. However, it is unknown if these cells are genuine memory cells or high-frequency naïve cell clones which bear overproduced TCR from bias in recombination/selection [15,46,47,48] or bias in homeostatic proliferation [49]. Also of interest was the alignment of multiple copy number clonotypes per donor with mean dextramer frequency hierarchy. For instance, the IMN response showed more donors with multiple copies of unique clonotypes in the auto-reactive cohort, and the FLC response showed more donors with multiple copies of unique clonotypes in the allo-reactive cohort. This aligned with dominance hierarchies from dextramer scanning. However, this correlation is speculative and requires additional investigation across more donors, antigens, and alleles. None-the-less, divergent mean T-cell frequencies were consistently observed when comparing auto-reactive versus allo-reactive individuals.

Gene usage analysis of the IMN and FLC repertoires showed significant differences in TRBV and *TRBJ* profiles between cohorts with a consistent preference for *TRBJ*1-1, TRBV2-1 TRBV2-7 in both allo-reactive systems. Allo-reactive repertoires also showed increased TRAJ usage in both systems. Additionally, auto-reactive T-cell repertoires exhibited a Gaussian distribution in CDR3 length whereas allo-reactive T-cell repertoires exhibited distorted CDR3 length distribution, with a tendency to encode shorter CDR3 loops. When examining TRBV/*JTRBJ* and TRAV/TRAJ pairing between auto-reactive and allo-reactive repertoires, allo-reactive repertoires showed a much more complex pairing network across both systems.

Physical-chemical property analysis showed that auto-reactive CDR3 in the IMN system were more hydrophobic, which was supported by a corresponding trend in increased GRAVY scores. This is of interest given hydrophobic CDR3 loops promote the development of self-reactive T-cells [50] and are enriched in both mouse [50] and human [51] T regulatory (T_reg_) cells. This suggests that auto-reactive IMN T-cells may be of the T_reg_ lineage. However, phenotypic analysis is required to validate this observation. Allo-reactive IMN CDR3 loops showed significant decreases in average Mw and monoisotopic Mw. This made sense given the IMN repertoire was strongly skewed to short CDR3 sequences previously seen in gene analysis. The FLC system did not follow this divergence in hydrophobicity and Mw; however, multivariate analysis of physical-chemical scores could clearly distinguish the auto-reactive repertoire from the allo-reactive repertoire. Of interest, both IMN and FLC allo-reactive CDR3 loops showed lower mean theoretical pI scores, meaning allo-reactive receptors tended to be more acidic in charge.

Diversity metrics of the IMN and FLC systems could clearly distinguish auto-reactive repertoires from the allo-reactive repertoires in multivariate analysis but trends in univariate analysis. Of note, the same Log_2_FC patterns were seen in both systems suggesting a possible general model for allo-reactive repertoires. Here, allo-reactive repertoires showed a general increase in diversity and consistency between individuals, meaning an increase in unique clonotypes, fewer unique multiple copy clonotypes and an increase in repertoire homogeneity between donors.

Motif analysis of CDR3 loops in the IMN and FLC systems could also clearly distinguish auto-reactive repertoires from the allo-reactive repertoires on multivariate analysis. Allo-reactive receptors in both systems showed increased motif pairing, showing that allo-reactive CDR3 loops exhibit more structural “options” when engaging an allo-reactive ligand. This finding aligns with increased TCR gene pairing observed in gene analysis and increased species observation in diversity analysis. While important, it must be said that these observations are preliminary, correlative and do not infer the cause of graft rejection. Future studies should be aimed at analysing additional sequences, additional antigens, additional donors for theory validation and patients for clinical validation.

Using single-cell sequencing, we have defined new auto-reactive and allo-reactive repertoires for the cancer antigens IMN and FLC, comprising 356 TCRαβ pairs in the dataset. These receptors can be used for further exploratory research and have translatable potential. Indeed, foreign-antigen-specific TCR show up to a 10,000-fold increase in affinity compared with self-antigen-specific TCR [52] and are in the range of pathogen-specific TCRs (KD = 0.1–10 µM) [53] which can bypass coreceptor help [54] and initiate cell death within five minutes [55]. Thus, the allo-reactive TCR identified here could apply to TCR gene transfer and adoptive cell therapy. Other potential therapies include “swapping” the allo-reactive peptide with an antagonist to prevent alloreactivity [54] or enhancing the affinity of the auto-reactive or allo-reactive TCR further using phage display [55]. These affinity enhanced receptors can be used as soluble drugs and are currently in clinical trials [56].

However, these receptors require functional validation before proceeding. Nevertheless, we are optimistic as the method used here has been published by our team numerous times, including functional validation in cancer-, Epstein-Barr virus-, cytomegalovirus- [39] and influenza-specific systems [57,58,59].

There are other variables to consider before TCR therapy, including “personal genetics” and “infectious history” which likely influence our personal T-cell repertoire makeup. Any two individuals can differ by up to 800 million single nucleotide polymorphisms (SNPs) between genomes [60]. This is termed personal genetics, and we currently do not understand how our personal genomes fully impact immune function. Another variable is our infectivity record which is defined by which pathogens we acquire over the course of life and at what age, termed our infectious history [61], which can impact on future immune challenges.

In summary, we show allo-reactive T-cell repertoires and be defined from auto-reactive repertoires through differences in enumeration and TCR architecture when examining gene usage, gene pairing, CDR3 length, CDR3 physical-chemical properties, diversity measures and CDR3 motif appearance and usage. These data increase the knowledge base of the mechanisms underlying T-cell alloreactivity and may lead to new adoptive therapies, TCR therapies and the rational design of TCR for therapeutic use in autoimmunity, infectious disease, and cancer.

## 4. Materials and Methods

### 4.1. Sample Preparation and Processing

Healthy donor blood was studied using protocols carried out in accordance with guidelines and regulations under QIMR Berghofer Medical Research Institute (QIMRB) (HREC P2058, approved 4/9/2014). Informed consent was obtained from all participants in the study. The study was performed according to the rules of the Declaration of Helsinki of 1975. Peripheral blood mononuclear cells were separated, as previously described [62]. Briefly, peripheral blood mononuclear cells (PBMCs) were isolated by Ficoll-Paque PLUS (GE Healthcare, Chicago, IL, USA) density gradient centrifugation and cryopreserved in R10 medium (RPMI-1640 containing 10% FCS) supplemented with 10% DMSO (Sigma-Aldrich, St. Louis, MO, USA). PBMC were stored at 10^6^ cells in a 1 mL vial and cryopreserved for batch analysis, which we have found increases the sensitivity of gene and protein detection and decreases experimental artefact. Eight-digit HLA typing was performed using NGS by BGI Genomics (BGI, Shanghai, China).

### 4.2. Flow Cytometry and Single-Cell Sorting

Cryopreserved PBMC were thawed in 37 °C R10 media with 1 uL/mL of DNAse 1 (Thermo Fisher Scientific, Waltham, MA, USA) and 0.3 nM of dasatinib (Axon Medchem, Reston, VA, USA). APC-conjugated dextramer was added for 30 min on ice using either HLA-0201-IMNDMPIYM-APC (IMN), HLA-0201-FLCMKALLL-APC (FLC), HLA-A0201-KVAELVHFL-APC (KVA) or HLA-A0201-YLEPGPVTV-APC (YLE) dextramer (Immunodex, Toronto, ON, USA). We next surface stained the cells with CD14-Pacific Blue, CD16-Pacific Blue, CD19-Pacific blue (all BioLegend, San Diego, CA, USA), LIVE/DEAD^®^ Fixable Aqua (Life Technologies, Carlsbad, CA, USA), CD3-PE and CD8-FITC (all BioLegend, San Diego, CA, USA) and sorting was performed as previously described [10]. Single cells were gated using FSC-W and FSC-H. Non-viable cells were dumped using LIVE/DEAD and CD14, CD16 and CD19. Dextramer^+^ cells were sorted twice at low flow rates on a FACSAria IIu sorter (BD, Franklin Lakes, NJ, USA) into chilled 96-well twin-tec PCR plates (Eppendorf, Hamburg, Germany). The cells were then centrifuged at 500× *g* for 2 min and immediately stored at −80 °C.

### 4.3. Single-Cell TCRαβ Amplification and Annotation 

As previously described [63], cDNA was generated using VILO (Thermo Fisher, Waltham, MA, USA) according to the manufactures’ protocol. A PCR was then performed to amplify the TCR locus followed by a then a nested PCR. Gels were run to confirm the amplification of positive products and cleaned for Sanger sequencing using QIAGEN plates (QIAGEN, Hilden, Germany). Post sequencing, TCRα and TCRβ transcripts were annotated using Bioedit and V-QUEST [64]. Random sampling and percentage usage graphs were generated in Excel 365 (Microsoft, 2019). Chord diagrams were generated in R using the *circlize* package [65]. CDR3 motifs were determined using GLIPH 2.0 [43]. Paired clones are listed in Appendix A.

### 4.4. Diversity Analysis

T-cell repertoire diversity metrics were determined from total counts from individual donors. The calculations comprise:



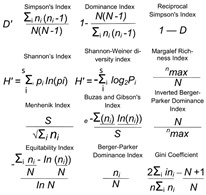



### 4.5. Physical-Chemical Properties

CDR3 loop hydrophobicity, GRAVY, average Mw, monoisotopic Mw and theoretical pI were determined using the Proteotypic Peptide Analyzing Tool (Thermo Fisher Scientific, Waltham, MA, USA).

### 4.6. Statistical Analysis 

TCR repertoire annotation was performed using IMGT/V-QUEST [64] and Excel 365 (Microsoft, USA). Post TCR gene alignment and CDR3 annotation, repertoires were then binned into individual dextramer^+^ repertoires or total pooled repertoires. Depending on the analysis, clonotypes were assigned to total repertoires (auto-reactive versus allo-reactive), abundance repertoires (where multiple clonal sequences were included) and appearance repertoires (where clonal sequences were decreased to one count and clonotypes randomly selected). Data was first checked for normality and lognormality using the D’Agostino-Pearson omnibus normality test and or the Shapiro-Wilk normality test. If the data failed these tests, a one sample t-test was performed with a Wilcoxon signed-rank test with mean calculated. Post quality control, a Mann-Whitney rank test, two-way ANOVA with Tukey’s multiple comparison test was performed using Prism V9.01 (Graphpad, USA). The *p*-value cut-offs are defined by * *p* < 0.05, ** *p* < 0.01, *** *p* < 0.001, **** *p* < 0.0001.

## Figures and Tables

**Figure 1 ijms-22-01625-f001:**
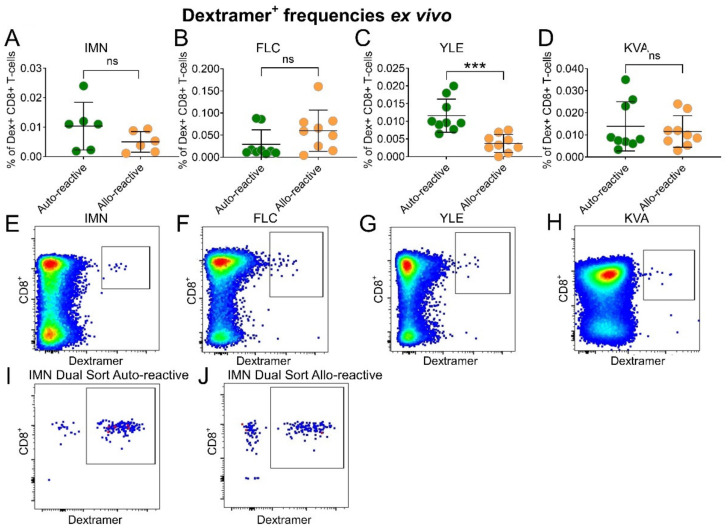
Different frequencies among dextramer populations in the blood for auto-reactive and allo-reactive T-cells. Dextramers include (**A**) IMN, (**B**) FLC, (**C**) YLE and (**D**) KVA. Frequency of CD8+ cells is shown on the y-axis, and dextramer frequencies from auto-reactive (green) and allo-reactive (orange) donors on the x-axis. Comparisons of IMN, FLC and KVA cohorts were not significant. Frequencies from the YLE-specific cohorts were significantly different, with the auto-reactive donors dominating. Examples of IMN (**E**), FLC (**F**), YLE (**G**) and KVA (**H**) dextramer staining ex vivo. Examples of dual (sequential) dextramer sorting from an (**I**) auto-reactive repertoire and (**J**) allo-reactive repertoire using IMN. ns, no significant; *** *p* ≤ 0.001.

**Figure 2 ijms-22-01625-f002:**
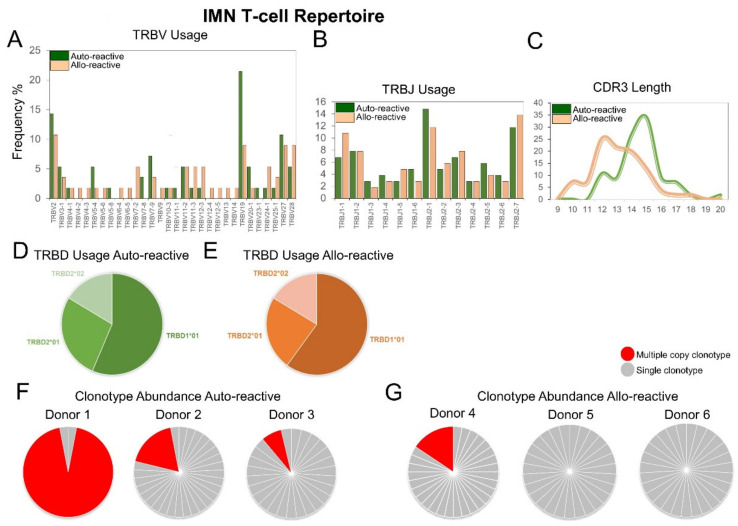
Different IMN-specific T-cell repertoires between auto-reactive donors and allo-reactive donors. Bar charts showing percentage use of (**A**) TRBV and (**B**) *TRBJ* preference from auto-reactive (green) versus allo-reactive (orange) donors. (**C**) Line chart comparing percentage use of CDR3 length. (**D**,**E**) Pie charts comparing percentage use of the TRBD genes. Pie charts showing clonotype frequencies in (**F**) three auto-reactive donors and (**G**) three allo-reactive donors. Each slice represents a unique clonotype from the repertoire of a single donor. Unique clonotypes with multiple copies are shown in red and unique clonotypes with a single copy are shown in grey. Percentages were determined by summing the total clonotypes found within a single donor.

**Figure 3 ijms-22-01625-f003:**
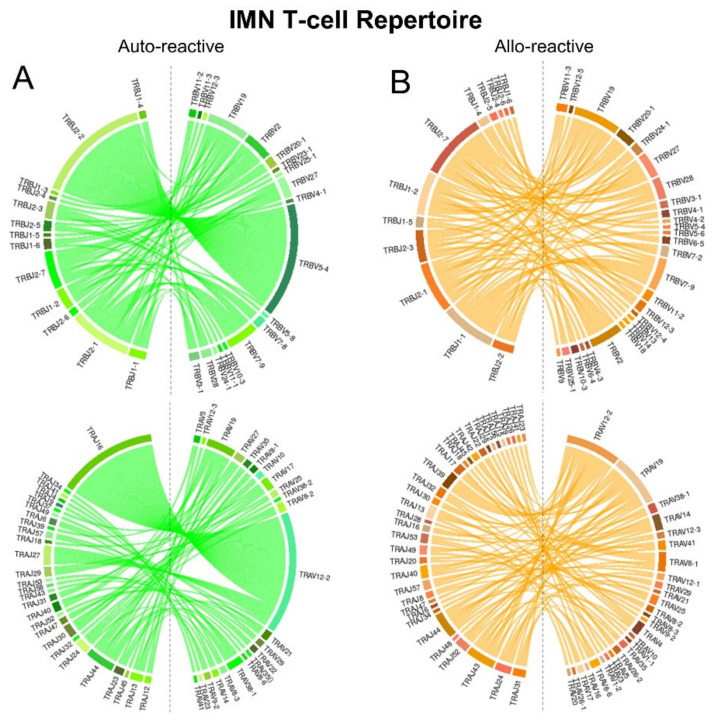
Chord diagrams of IMN-specific (**A**) auto-reactive (green) and (**B**) allo-reactive (orange) repertoires showing TRBV-*TRBJ* and TRAV-TRVJ pairings. The size of the connections represents an increased frequency of appearance.

**Figure 4 ijms-22-01625-f004:**
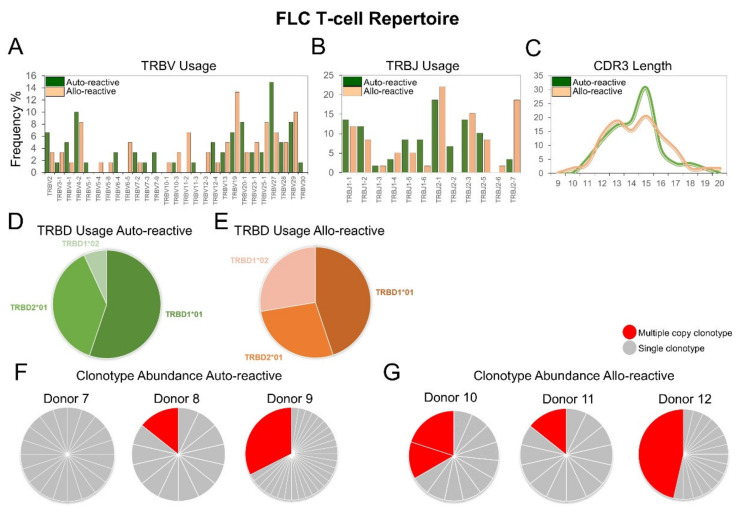
Different FLC-specific T-cell repertoires between auto-reactive and allo-reactive donors. Bar charts showing percentage use of (**A**) TRBV and (**B**) *TRBJ* preference from auto-reactive (green) versus allo-reactive (orange) donors. (**C**) Line chart comparing percentage use of CDR3 length. (**D**,**E**) Pie charts comparing percentage use of the TRBD genes. Pie charts showing clonotype frequencies in (**F**) three auto-reactive donors and (**G**) three allo-reactive donors. Each slice represents a unique clonotype from the repertoire of a single donor. Unique clonotypes with multiple copies are shown in red and unique clonotypes with a single copy are shown in grey. Percentages were determined by summing the total clonotypes found within a single donor.

**Figure 5 ijms-22-01625-f005:**
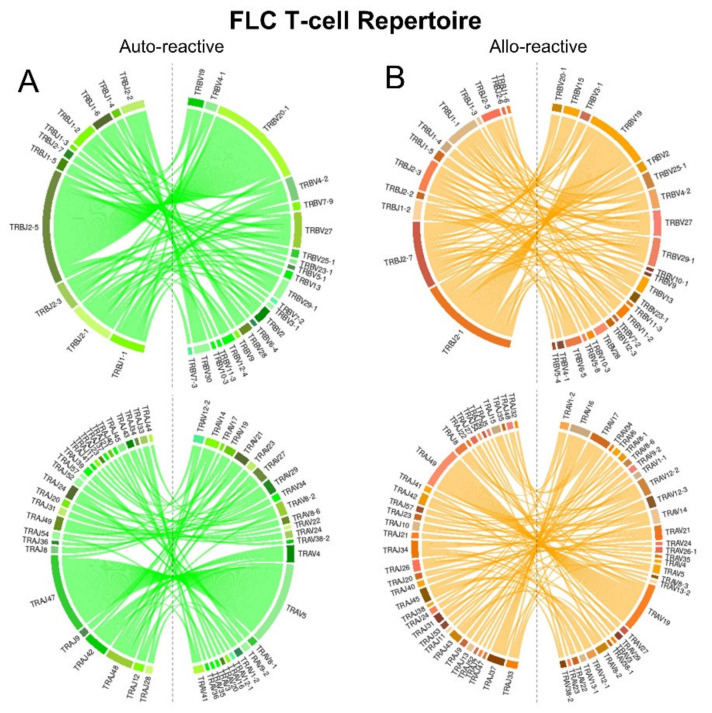
Chord diagrams of FLC-specific (**A**) auto-reactive (green) and (**B**) allo-reactive (orange) repertoires showing TRBV-*TRBJ* and TRAV-TRVJ pairings. The size of the connections represents an increased frequency of appearance.

**Figure 6 ijms-22-01625-f006:**
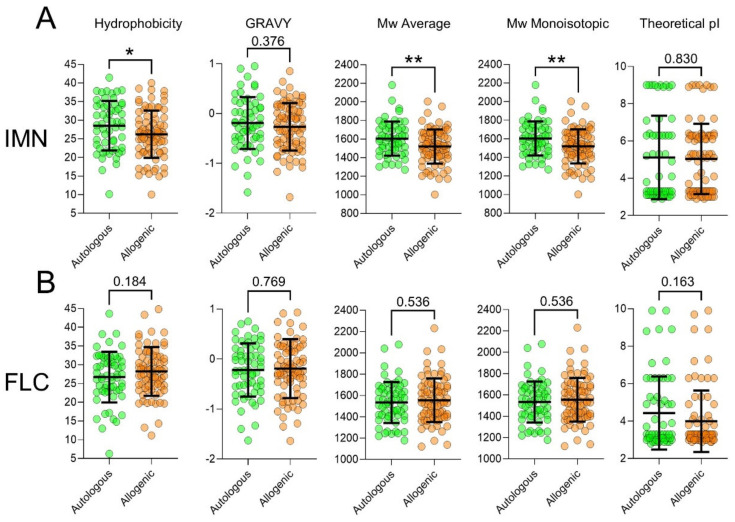
Physical-chemical properties analysis of CDR3 loops between auto-reactive and allo-reactive TCRs. (**A**) IMN CDR3 (green) and (**B**) FLC CDR3 loops (orange) were compared by hydrophobicity, GRAVY, Mw average, Mw monoisotopic and theoretical pI scores. TCR were pooled and measured by appearance. * *p* ≤ 0.05, ** *p* ≤ 0.01.

**Figure 7 ijms-22-01625-f007:**
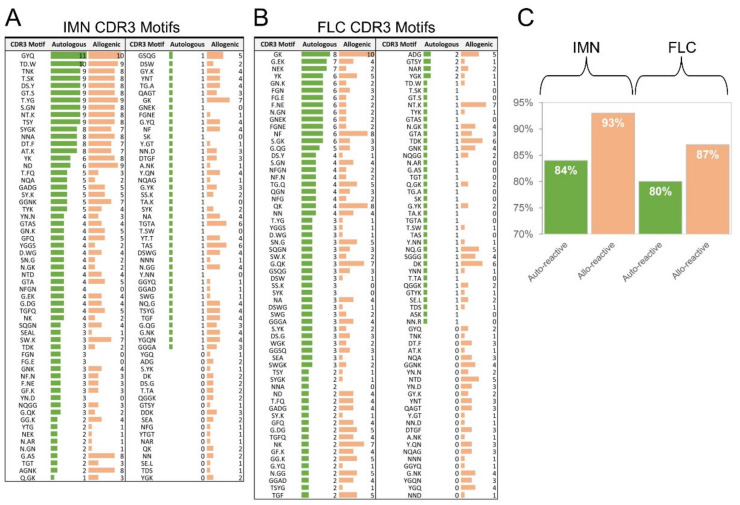
CDR3 motif analysis between auto-reactive and allo-reactive TCRs. (**A**) IMN and (**B**) FLC CDR3 loops found by motif searching. (**C**) Total percentage of IMN and FLC CDR3 loops found by motif searching. Auto-reactive (green) and allo-reactive (orange) sequences are shown as are the number of sequences encoding the motif.

**Table 1 ijms-22-01625-t001:** (A), IMN T-cell repertoire; (B) FLC T-cell repertoire.

**A: IMN T-cell repertoire**		
**Diversity Metric**	***p* Value**	**Log_2_FC**
Biodiversity	0.38	−0.15
Simpson’s index	0.41	−5.45
Dominance index	0.41	0.51
Reciprocal Simpson index	0.75	−0.54
Shannon index	0.39	0.56
Menhinick index	0.37	0.57
Buzas and Gibson’s index	0.35	0.34
Equitability index	0.40	0.43
Berger-Parker Dominance index	0.35	−2.41
Inverted Berger-Parker Dominance index	0.17	1.63
Margalef Richness index	0.38	0.59
Gini coefficient	0.34	−2.46
**B: FLC T-cell repertoire**		
**Diversity Metric**	***p* Value**	**Log_2_FC**
Biodiversity	0.99	0.00
Simpson’s index	0.73	−0.71
Dominance index	0.73	0.10
Reciprocal Simpson index	0.65	−0.65
Shannon index	0.90	0.05
Menhinick index	0.81	0.10
Buzas and Gibson’s index	0.90	0.06
Equitability index	0.77	0.08
Berger-Parker Dominance index	0.84	−0.24
Inverted Berger-Parker Dominance index	0.56	0.81
Margalef Richness index	0.99	0.01
Gini coefficient	0.92	−0.14

## Data Availability

The TCR data is available at https://vdjdb.cdr3.net.

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
