# Peer review of "Genetic Bias, Diversity Indices, Physiochemical Properties and CDR3 Motifs Divide Auto-Reactive from Allo-Reactive T-Cell Repertoires"

_ijms, 2021, doi:10.3390/ijms22041625_

Round 1

Reviewer 1 Report

Revisions are appropriate

Author Response

Thank you.

Reviewer 2 Report

The authors never define what they mean by autologous and allogenic. In the beginning of the results section, they should include a statement that HLA-A2-dextramers were used to stain PBMC from HLA-A2+ and HLA-A2- individuals. And then be consistent with the labeling throughout. They begin with using autologous and allogenic and then switch to HLA-A2+ and HLA-A2- which is very difficult to distinguish in the figures. 

The authors begin by looking at 4 peptides and show that only 1 of them does the precursor frequency differ. But, they then proceed to look at only 2 of the peptides in detail with no explanation of why only those 2 peptides which exclude the one peptide that showed a difference in precursor frequency.

The figures would benefit from better labeling.  For example, we have no idea where Fig1E-H come from. In addition, Fig1I-J are supposed to show dual dextramers but are still labeled as CD8 vs Dextramer. The other figures also need labeling and revisions.

The entire document needs editing for language and consistency. The are numerous erroneous, misleading and incomplete sentences.

In section 2.2 (which is actually 2.3), the authors talk about clonotype expansion, but it is unclear what they mean by this. How were these clonotypes expanded? If they mean abundance of individual clonotypes then this should be changed. However, if this is the case, it is unclear how 90% of the PBMC from one donor are the same clone. If this is within the peptide specific population, that needs to be stated. 

Figure 2F-G are labeled as clonal expansion as is Figure 3. What is the difference between these figures? Why are there 2 figures for the same dataset. 

Similarly, figure 4 and 5 seem to be the same thing. Figure 4 provides more information so Figure 5 is irrelevant and should be deleted. 

The above comments then apply to the same duplicate figures in Fig6-9 for the second peptide.

Figure 10 is a mix of the 2 different peptides and should be labeled with their respective peptides or separated to distinguish between them.

The table provides a lot of information, but is difficult to sort through. I would recommend splitting into 2 tables, one for each peptides, and adding another column to each to highlight variances either with p-values or differences.

The results section is really lacking in descriptions of the data presented in the figures. Some of this description is in the discussion section and should be moved to the results section to make it more understandable as to what has been done and what the data shows.

The overall conclusion seems to that there is no one factor that determines this difference in recognition but that several of these factors should be further investigated. It would be helpful for the authors to conclude with a synthesis of what these factors that they point out could collectively mean for the difference in recognition between autologous and allogenic receptors. This would increase the impact that this study could have.

Author Response

We thank Reviewers 2 for their time reading this manuscript and for their c
